# Fluctuating Asymmetry in the Polymorphic Sand Cricket (*Gryllus firmus*): Are More Functionally Important Structures Always More Symmetric?

**DOI:** 10.3390/insects13070640

**Published:** 2022-07-16

**Authors:** Matthew R. Whalen, Krista J. Chang, Alexandria B. Jones, Gabriel Rivera, Amy M. Worthington

**Affiliations:** Department of Biology, Creighton University, 2500 California Plaza, Omaha, NE 68178, USA; mrwhalen@umich.edu (M.R.W.); kristachang@creighton.edu (K.J.C.); alliejones@creighton.edu (A.B.J.)

**Keywords:** fluctuating asymmetry, directional asymmetry, symmetry, polyphenism, life history, cricket, Orthoptera, Gryllidae, *Gryllus*

## Abstract

**Simple Summary:**

Asymmetry in bilateral structures occurs when animals experience perturbations during development. This fluctuating asymmetry (FA) may serve as a reliable indicator of the functional importance of a structure. For example, locomotor structures often display lower levels of FA than other paired structures, highlighting that selection can maintain symmetry in traits important for survival or reproduction. Species that have multiple distinct morphs with unique behaviors and morphologies represent an attractive model for studying the relationship between symmetry and function. The sand field cricket (*Gryllus firmus*) has two separate morphs that allow us to directly test whether individuals maintain higher levels of symmetry in the structures most vital for maximizing fitness based on their specific life strategy. Longwing (LW) individuals can fly but postpone reproduction until after a dispersal event, whereas shortwing (SW) individuals cannot fly but begin reproducing in early adulthood. We quantified FA across a suite of key morphological structures indicative of investment in growth, reproduction, and flight capability for males and females across the morphs. Although we did not find significant differences in FA across traits, as predicted, locomotor compensation strategies may reduce selective pressures on symmetry or developmental patterns may limit the optimization between trait form and function.

**Abstract:**

Fluctuating asymmetry (FA) may serve as a reliable indicator of the functional importance of structures within an organism. Primary locomotor structures often display lower levels of FA than other paired structures, highlighting that selection can maintain symmetry in fitness-enhancing traits. Polyphenic species represent an attractive model for studying the fine-scale relationship between trait form and function, because multiple morphs exhibit unique life history adaptations that rely on different traits to maximize fitness. Here, we investigated whether individuals of the wing polyphenic sand field cricket (*Gryllus firmus*) maintain higher levels of symmetry in the bilateral structures most vital for maximizing fitness based on their specific life history strategy. We quantified FA and directional asymmetry (DA) across a suite of key morphological structures indicative of investment in somatic growth, reproduction, and flight capability for males and females across the flight-capable longwing (LW) and flight-incapable shortwing (SW) morphs. Although we did not find significant differences in FA across traits, hindwings lacked DA that was found in all other structures. We predicted that functionally important traits should maintain a higher level of symmetry; however, locomotor compensation strategies may reduce the selective pressures on symmetry or developmental constraints may limit the optimization between trait form and function.

## 1. Introduction

Organismal fitness depends on a strong matching of morphological structures with their usage in daily life, with more adaptive phenotypes conferring enhanced survival and reproduction. To fine-tune this relationship between form and function, selection acts on fixed genetic differences to alter morphology across generations; however, trait plasticity is equally important for rapidly optimizing fitness to changing environmental conditions experienced by an individual. Polyphenic species represent an attractive model for studying such trait plasticity [1,2] because multiple morphs exist within a single species and each morph exhibits unique life history adaptations resulting from a combination of genetic and environmental factors [3,4,5]. Polyphenic morphs can differ in nutritional allocation (crickets: [6]), structural development (stag beetles: [7]), gene expression (termites: [8]; horned beetles: [9]), and behavior (beetles: [10]; butterflies: [11]), with each morph expressing a phenotype that matches the selective environment it inhabits. Despite arising from the same general body plan, differing developmental trajectories can produce significant variations in the size, shape, and functionality of morphological structures between the morphs [12]. Additionally, in cases where morphs comprise both sexes, males and females face distinct selective pressures and differentially allocate resources to sexually selected traits [13,14]. The result is that within a single polyphenic species, multiple distinct groups exist with specialized morphologies to maximize fitness based on their respective developmental background, thereby providing a unique opportunity to assess the impact that different sexes and life history strategies have on the growth and maintenance of functionally important structures at a fine scale.

Although functional variation in the size and shape of morphological structures within polyphenic species is well studied [14,15,16], the associated parameter of symmetry has received considerably less attention [17]. Fluctuating asymmetry (FA) is defined as random deviations from perfect symmetry in bilateral structures [18], and is often used as a measure of an individual’s ability to buffer against environmental perturbation (i.e., developmental stability: [18,19,20,21]. FA may also serve as a reliable indicator of the functional importance of structures within an organism [18,22], given that FA has a heritable component upon which evolution can act [23,24,25], and that bilaterally paired structures may require a high degree of symmetry to remain functional. To date, this hypothesis has primarily been tested in vertebrates and among species occupying unique ecological niches. In both anurans (Bufonidae and Ranidae: [17]) and turtles (Emydidae: [26]), hindlimbs play a greater role in locomotion (e.g., greater force production) than forelimbs; concordantly, they exhibit lower levels of FA. In addition to finding that more functionally important structures (i.e., hindlimbs) display lower levels of FA, Rivera and Neely [26] also showed that low levels of FA have been canalized in the hindlimbs, in contrast to patterns in the forelimbs, where rates of evolutionary change are approximately sixfold greater. Importantly, asymmetry of bilateral structures not only affects trait functionality, but can strongly be related to fitness. For example, cliff swallows [27], kelp perch [28], and woodmice [29] that exhibited greater FA in key locomotor traits (i.e., wings, pectoral fins, and legs, respectively) suffer increased rates of mortality compared with individuals displaying lower levels of FA. These studies provide compelling evidence that FA serves as a reliable indicator of the functional importance of a trait within organisms, and that selection can act on FA to maintain the symmetry of functionally important traits.

Few studies have assessed FA across polyphenic groups within a single species. Crespi and Vanderkist [30] investigated the levels of FA in functional and vestigial traits in polyphenic gall thrips (*Oncothrips tepperi*), and reported that the wings of the dispersal morph exhibited significantly lower levels of FA than the wings of the non-flighted soldiers [30]. These results support the hypothesis that levels of FA in a trait increase when selection for trait functionality is relaxed. Stag beetles (*Prosopocoilus inclinatus*) also exhibit polyphenism, with clear differences in mandible size and function between sexes and across life stages. Okada et al. [31] found that larval and adult female stag beetles exhibit strong directional asymmetry (DA, when a paired structure is typically larger on a specific side for the population as a whole), such that their mandibles cross and function as scissors for cutting through wood fibers. In contrast, adult males possess enlarged mandibles that are primarily used as pinching weapons for male combat and lack directional asymmetry. In both studies, the traits investigated were limited to the traits that exhibited the most extreme functionality differences—it remains unclear whether additional structures that differ in functionality between the morphs and/or sexes exhibit similar patterns in either the canalization of symmetry or the relaxation of developmental stability.

To bridge this gap, we examined FA in the bilateral structures of the polyphenic sand field cricket, *Gryllus firmus* Scudder 1902. Field crickets, including *G. firmus*, exhibit a wing polymorphism comprising longwing (LW) and shortwing (SW) individuals. The flight-capable LW morph has fully formed hindwings supported by large flight muscles allowing for dispersal, whereas the flightless SW morph has underdeveloped hindwings and flight muscles, instead exhibiting enhanced early reproductive potential [32,33]. Wing dimorphism is therefore typically characterized in this species as a physiological trade-off, in which resources are preferentially allocated between flight capability [34,35] or fecundity [32,36]. Unlike some wing polymorphic insects, flight-capable *Gryllus* do not dealate their hindwings after dispersal, but rather histolyze their flight muscles prior to reallocating resources into reproduction [37]. In addition to differences between these wing morphs, sexual selection plays an important role in the reproductive strategies of males and females in *Gryllus* crickets. Males use specialized calling structures on their wings to attract mates [38], and females use a collection of auditory [39] and chemical cues [40] from males to discriminate between potential sires. Given these distinct life history strategies of the wing morphs and sexes, individuals likely rely on different morphological structures to maximize their fitness, and the strength of selection acting on the functionality of those structures could vary significantly between the morph × sex groups.

In this study, we tested the hypothesis that individuals maintain symmetry in structures most vital for maximizing fitness based on their specific life history strategy. To test this hypothesis, we quantified FA across a suite of key morphological structures indicative of investment in somatic growth (mouthparts: mandibles and maxilla), reproductive investment (forewings and tympanum), terrestrial locomotion (legs: fore/hind tibia and femur), and flight capability (hindwings and tympanum) for males and females across both LW and SW morphs. Additionally, percentage body fat was employed as a measure of individual quality to assess whether lower-quality individuals exhibited higher levels of FA. Finally, we quantified the shape of the complex forewing (i.e., tegmina) structures in males responsible for producing the sexually selected mating calls using geometric morphometrics and calculated FA from these multivariate data to test for differences between SW and LW males. We predicted that the structures most essential for fitness would exhibit the lowest degree of FA across all the structures we quantified, and that the most functionally important traits in each morph × sex group would exhibit lower levels of FA relative to those same traits in the alternative morph or sex. Specifically, we predicted that across both sexes, LW morphs would show lower levels of FA in the primary flight structure (i.e., hindwings) relative to the SW morph. Finally, although each morph and sex exhibits unique life history strategies, we predicted to find no difference in FA between the morphs in structures that are important to survival for all individuals, including terrestrial locomotion (e.g., foreleg and hindleg structures) or nutrient acquisition (e.g., maxilla).

## 2. Methods

### 2.1. Animal Rearing

We used *Gryllus firmus* crickets from a laboratory population originally started by Dr. Anthony Zera at University of Nebraska—Lincoln. This population was derived from wild-caught individuals collected in Gainesville, FL, in 1995, and two lines have been continually artificially selected for either the flight-capable longwing morph (LW) or the flight-incapable shortwing morph (SW; [41]) for approximately 90 generations. Despite being reared under laboratory conditions, these genetic populations still display significant life-history differences in their metabolic capacities for energy production [42], response to environmental stressors [43], immunocompetence [44,45], and gene expression [46].

Our large general populations (each maintained with >2000 individuals) were reared in 85 L clear plastic bins with ventilated lids. They were supplied stacks of egg cartons for structure, fed Special Kitty Premium cat food ad libitum, and provided large cotton-plugged water vials for moisture. All bins were housed in an environmentally controlled room at 26–28 °C, 70–80% relative humidity, and on a 12:12 h light:dark cycle. Crickets in this experiment were reared collectively with approximately 50–75 mixed-sex nymphs of their own LW or SW line starting at hatching in the fall of 2017. Upon molting into their penultimate instar, a time when individuals can be reliably sexed, male and female crickets were separated into single-sex bins. Nymphs were checked daily for eclosion, and a minimum of forty newly eclosed adults of each sex–morph group (e.g., F–LW, F–SW, M–LW, and M–SW) were removed from the communal bins and reared individually in 250 mL plastic cups with ad libitum food and water for 7 days. Crickets were then euthanized and stored at −20 °C until analyzed. The final sample sizes of crickets collected are as follows: F–LW = 49, F–SW = 36, M–LW = 42, M–SW = 34.

### 2.2. Data Collection

We used the pronotum length of each cricket as a proxy for each specimen’s overall size [47]. Paired forelegs (including tympana), hindlegs, mandibles, and maxillae were carefully dissected from each specimen at the body attachment joint and temporarily mounted using modeling clay to ensure that all structures were completely parallel to the microscope lens to avoid parallax. These traits were then photographed using a Leica IC90-E camera mounted on a Leica M80 stereoscope. Hindwings of individuals were removed at the attachment to the thorax and mounted in nail polish between two large glass slides (76 mm × 50 mm). The ventral side of each wing was spread onto nail polish using forceps and allowed to set for 20 s; then, a second layer was applied to the dorsal side of the wing. Another slide was then added on top of the wing and held in place under pressure for 90 s to allow the nail polish to dry completely with the wing fully outstretched. Images were taken immediately using a Nikon D3400 camera (f/8; 25 mm; ISO 100) mounted at 60 cm above an LED lightbox (Voilamart A2).

All linear and area (used only for tympanum) measurements were taken using ImageJ software (v1.31; https://imagej.nih.gov/ij/, accessed on 1 June 2022). Diagrams of the raw linear measurements for each external structure are presented in Figure 1. For each structure analyzed, one researcher took all photographs and measurements to avoid inter-observer variation. We repeated measurements three times in a random order of individuals and R/L structure without reference to previous measurements to obtain an estimate of intra-observer measurement error.

Forewings were removed by cutting through the articular sclerites at the attachment point to the thorax. After removal, each wing was pressed gently between two large glass slides. The ventral side of each wing was imaged using a digital camera (Leica IC90E) mounted on a stereo-dissecting microscope (Leica MZ8, 0.6× objective) with under-stage lighting. All photos were taken at full brightness to ensure that venation was maximally visible. Before landmark digitization, photographs of left wings were horizontally flipped so that left and right wings could be directly compared rather than being mirrored structures. For females, we selected 9 landmarks that provided a reliable outline of the wing, and for males, we selected 17 landmarks to analyze both the outer margin of the wing and the principal structures involved in acoustic courtship signaling (as shown in Figure 2). Each forewing was digitized three times to account for measurement error using TPSdig software (v1.07; http://life.bio.sunysb.edu/morph/, accessed on 1 June 2022).

Finally, we quantified the percentage body fat of each cricket included in our study to identify whether levels of FA correlated with an individual’s energetic reserves. Crickets were dried at 60 °C for 24 h and weighed to the nearest 0.01 mg using an electronic balance to determine their lean dry mass. Body fat was then extracted using petroleum ether (Fisher Scientific, Hampton, NH, USA) reflux in Soxhlet apparatus for 12 h. Individuals were again dried at 60 °C for 24 h and then reweighed to obtain their lean dry mass. Body fat content (mg) was obtained by subtracting the lean dry mass from the dry mass, and the percentage body fat was then calculated by dividing body fat content by that individual’s lean dry mass.

### 2.3. Statistical Analysis

#### 2.3.1. Univariate Data

Asymmetry in bilateral structures is measured as the difference between the left and right sides (*R*–*L*) of a structure for an individual [18]. Within the sample population, these values are expected to produce a normal distribution, where the unsigned magnitude of random deviations from that mean, |*R* – *L*|, represents the level of FA for each individual and the mean signed difference between the right and left sides, ($\bar{R-L}$), corresponds to directional asymmetry (DA) for the population We used a modification of this general equation to isolate FA values, as described below. In addition, the magnitude of the difference between sides |*R* – *L*| is often correlated with the size of the structures being measured, with increases in size generally yielding larger differences between sides (Palmer & Strobeck 1986). To account for the differences in trait size between individuals, and thus facilitate comparisons between sex × morph groups, we standardized each calculated difference (*R* – *L*) by dividing it by the average size of the structure examined [17,18,26,48,49]. This dimensionless index of proportional difference, hereafter referred to as ‘individual measurement’, is the most common transformation used to correct for variation in trait size. To ensure that this transformation did in fact remove the size-dependence of FA data without inducing additional patterns, we conducted regression analyses for unscaled and scaled side-difference magnitudes against the average structure size [17,26].

The aim of our study was to examine normal deviations from symmetry that are consistent with patterns of developmental instability. To do this, it was important to maintain a high level of variability in our data, but also to set an upper threshold for extreme differences that might be indicative of developmental abnormalities. As such, for each of the 32 datasets (two sexes × two morphs × eight structures), we excluded individual measurements more than three times greater than the interquartile range below or above the first and third quartile, respectively (<Q_1_ − 3.0 × IQR or >Q_3_ + 3.0 × IQR; see Rivera and Neely [26]). Once outliers were removed, we tested for differences in size for each measured structure (i.e., the average of the left and right sides), using a two-way ANOVA (sex × morph). Following each ANOVA, pairwise comparisons of all sex-by-morph groups were conducted using a Tukey post hoc test. Statistical analyses were conducted using R (v3.6.1; R Core Team 2019) unless otherwise noted.

We calculated corrected FA values by removing the effects of DA using the following equation:$${\mathrm{FA}}_{\mathrm{corrected}}=\sqrt{\left|\;\left[\frac{R-L}{0.5\times \left(L+R\right)}\right]-\left[\frac{\sum \left(\frac{R-L}{0.5\times \left(L+R\right)}\right)}{N}\right]\;\right|}$$
where the first component is the individual measurement, $\left[\frac{R-L}{0.5\times \left(L+R\right)}\right]\;$, and the second component, $\left[\frac{\sum \left(\frac{R-L}{0.5\times \left(L+R\right)}\right)}{N}\right]$, represents the mean size-corrected signed difference for the sample (i.e., DA). The magnitude of the difference between these two components was then square-root-transformed to produce a normally distributed sample. We tested for normality of each dataset using the shapiro.test function in R, then tested for significant DA using a two-tailed one-sample *t*-test to determine if the mean differed significantly from zero.

FA_corrected_ values were calculated for all 32 datasets, after which we tested whether sex and/or morph influenced FA using a two-way ANOVA (sex × morph). Following each ANOVA, pairwise comparisons of all sex × morph groups were conducted using a Tukey post hoc test. We also tested whether FA_corrected_ was correlated with the quantified body fat percentage, because individuals with larger fat stores had more energetic resources available for the growth, development, and maintenance of anatomical structures. For each of the eight structures analyzed, we conducted a linear regression analysis between FA_corrected_ and body fat percentage for all sex × morph groups pooled. Additionally, we included FA_corrected_ for all eight structures in a single PCA and performed linear regressions between the first three PCs and body fat percentage for all sex × morph groups pooled.

For each dataset, we also conducted a two-way mixed-model ANOVA (Side*Id), with ‘Sides’ as fixed factors, ‘Id’ (i.e., individuals) as random factors, and ‘replicates’ as the error term. This analysis provided a test of whether between-sides variation is significantly greater than measurement error—a significant interaction variance (MS_Side*Id_) is a prerequisite for meaningful tests of FA [18,48]. The ANOVA also provided a test of directional asymmetry (DA = MS_Side_) and for the quantification of measurement error (ME3: %ME = [MS_error_/MS_Side*Id_] × 100; [49]). Additionally, MS_Side_/MS_Side*Id_ provides the ratio of DA:FA, allowing for direct comparison of the magnitudes of these two types of asymmetry. Finally, this analysis also allowed for the calculation of the asymmetry index FA10 ([MS_Side*Id_ − MS_error_]/# replicate measurements; [48]), which removes the variance associated with measurement error and DA, but has the limitation of not allowing for the correction of size variation [48,49]. These ANOVAs were performed using IBM SPSS Statistics 24.

#### 2.3.2. Multivariate

We used geometric morphometric methods to analyze the asymmetry of shape of the forewings for both males and females. Sexes were analyzed separately because different landmarks are present in the two sexes. As we had already mirrored wing images prior to digitization, we first aligned coordinates within each dataset using a generalized Procrustes analysis (GPA; [50,51]), which removes all non-shape information (translation, scale, rotation) from the dataset. We then conducted a Procrustes ANOVA (function “bilat.symmetry” in the R package “geomorph” v4.0; [52]) to determine the effects of measurement error on FA and test for the significance of FA, an analysis analogous to the mixed-model ANOVA used for our univariate data [53]. Our goal was to test for differences in shape asymmetry between morphs within each sex; therefore, it was necessary to assign values of FA to each individual. To do this, we calculated the pairwise Procrustes distances between all wings in the dataset and extracted the calculated distance between the left and right wing of each individual. This single value was then used as a proxy for individual FA [53] and was analyzed using a one-factor ANOVA, testing the effect of morphotype.

## 3. Results

Asymmetry data were analyzed for 161 specimens (N for each sex–morph: F–LW = 49, F–SW = 36, M–LW = 42, M–SW = 34). Not all variables were able to be measured for each individual. Across all eight linear variables, 35 data points were excluded as extreme outliers, representing approximately 3% of the total potential data points (N = 1195). For all variables except for hind femur length and hindwing length, the number of exclusions ranged from one to four. For hind femur length and hindwing length, there were eight and eleven exclusions, respectively. In addition, two individuals were excluded from the GMM dataset as outliers. These exclusions represent extreme deviations from the normal development of the measured structures due to factors such as difficulty molting or injury incurred during a juvenile instar.

Pronotum length, used as a proxy for body size, was available for 158 of the 161 specimens, with one value missing for M–LW and two values missing for M–SW. Females of both morphs (Mean ± SD: F–LW = 4.36 ± 0.27; F–SW = 4.28 ± 0.28) were larger than males of both morphs (Mean ± SD: M–LW = 3.94 ± 0.28; M–SW = 3.88 ± 0.37). Results of a two-factor ANOVA on pronotum length (Appendix A) found a significant effect of sex (*p* < 0.001), but not morph (*p* = 0.146) or the interaction term (*p* = 0.755). A comparison of linear size for the other eight bilateral traits is found in Figure 3. Tukey tests (Appendix A) showed that males differed significantly from females irrespective of morph (*p* < 0.001), whereas intra-sex comparisons of morphs were not significantly different (*p* > 0.58).

We conducted linear regressions on raw and size-adjusted FA values to examine the pattern of size dependence of our data and to ensure that our transformations were appropriate. Of the thirty-two regressions performed (four morphs × eight variables), five displayed significant effects of size. Our transformations, which allowed us to examine proportional FA, successfully removed the significant size effects from four of these and changed the level of significance for the fifth (M–LW mandible: raw to size-adjusted, <0.001 to 0.024; see Appendix A), indicating that our transformation of raw FA to size-adjusted FA did remove significant size-effects from our data. In addition, corrected FA values (i.e., size-adjusted FA with DA removed) fitted a normal distribution for all 32 datasets (Shapiro–Wilk normality test: *p* > 0.065; Appendix A). 

The results of our two-way mixed-model ANOVAs found that between-sides variation was significantly greater than measurement error for all 32 tests (*p* < 0.001) (Table 1). The mean measurement error for all 32 datasets was 5.7% (range = 0.2–20.5%). The results of MS_Side_ from the mixed-model ANOVA indicate considerable DA across datasets. Table 1 reports the FA10 values; however, these values are influenced by variation in body size between morphs, and are thus difficult to interpret.

Significant DA was pervasive throughout most datasets (*p* < 0.01 for 27 of 32 datasets: Table 2). Only the hind femur of M–SW and all hindwing datasets lacked DA (*p* > 0.057). Additionally, of the 27 significant datasets, all but the front tibia and tympanum datasets (N = 8) had larger structures on the left side of the body (as indicated by negative DA values in Table 2).

No global pattern for corrected FA values was detected (Table 2; Figure 4). Two-factor ANOVAs (sex × morph) conducted on each variable also found limited significance. Across the eight tests, we found four significant parameters (Appendix A): interaction terms for front femur (*p* = 0.013), hind femur (*p* = 0.022), and tympanum (*p* < 0.001), and sex for maxilla (*p* = 0.007). Tukey post hoc tests found eight pairwise differences: for hind femur and tympanum, M–SW had significantly higher FA than all other groups (*p* < 0.048); for front femur and maxilla, F–LW had significantly higher FA than M–LW (*p* < 0.045; Appendix A).

To determine whether the level of nutritional resources influenced developmental stability, we examined the relationship between the percentage body fat and FA across all 32 datasets. We found only one significant relationship: a positive correlation between the percentage body fat and FA for the maxilla of F–LW (df = 43, t = 3.33, r = 0.45, *p* = 0.002; Appendix A).

Our analysis of shape (GMM MANOVA) indicated that for all four groups, differences in shape between forewings was significantly greater than differences associated with measurement error (Individual × Side = *p* < 0.001, Appendix A). Additionally, although DA was not present in the forewings of either female morph (*p* > 0.673), it was present in the forewings of both male morphs (*p* < 0.008, Appendix A). ANOVAs of the Procrustes distance between the forewings of each individual (used as a measure of FA) indicated that there was no significant difference in FA between morphs in males (df = 1, F = 1.87, *p* = 0.177) or females (df = 1, F = 1.98, *p* = 0.165).

## 4. Discussion

We analyzed fluctuating asymmetry (FA) of functionally important traits in a polyphenic species to test the hypothesis that morphs and sexes maintain greater symmetry (i.e., lower FA) in structures most vital for maximizing fitness based on their specific life history strategy. In contrast to our predictions, we found no statistically significant evidence that the longwing (LW) dispersal morph had lower FA in hindwings used for flight relative to the non-dispersing shortwing (SW) morph. Furthermore, we found that male SW individuals had higher FA in their hind femur and tympanum relative to the LW morph and females. Interestingly, we found evidence of directional asymmetry (DA) for nearly all linear traits, with the exception of the hindwings. We found no significant FA in the forewing shape; however, males, but not females, exhibited DA in these structures.

Asymmetry in locomotor structures has been shown to reduce performance in a broad range of animal taxa [54,55,56,57,58,59]. Specifically, non-symmetric flight structures can produce asymmetric aerodynamic forces (e.g., lift and drag), requiring complex compensation strategies and increasing the metabolic costs for flight [60]. We predicted that the flight-capable LW morph would show greater levels of symmetry in their primary flight structure (i.e., hindwings) relative to the flight-incapable SW morph due to the functional constraints of a flighted dispersal. Highlighting the extent that trait investment strategies diverge in this polyphenic cricket, LW morphs of both sexes had hindwings greater than twice the length than those of SW individuals. We found partial support for our hypothesis, in that the hindwings were the only bilateral structure lacking significant DA. We did not, however, find evidence that FA was lower in the hindwings compared with other bodily structures where symmetry was less likely to be canalized (Figure 4), nor did the hindwings of LW crickets differ significantly in FA compared with those from SW individuals. It is important to note that all moderate outliers retained in the analysis (N = 5) belonged to the SW morph. Thus, although the majority of individuals of both morphs maintained relatively equivalent levels of FA, when FA deviated from the mean, SW morphs tended to have more extreme FA than LW individuals.

There are a number of potential explanations for the lack of significant differences in FA in hindwings across groups. First, although there is considerable evidence that wing asymmetry has a negative functional impact on flight in birds [59], such a link in insects has not been established. Unlike birds, the delicate membranous wings of insects often incur irreparable asymmetric damage with age [61]. Considering that insects must live with this damage, there may be strong selection for the development of compensatory mechanisms (e.g., modified flight kinematics) that prevent reductions in flight performance [62]. In fact, studies investigating experimentally induced wing damage found that asymmetries up to a 10% difference in wing area may cause few, if any, significant consequences on performance [60,63]. Even at asymmetries >10%—well beyond the typical level of FA—the primary impacts are on complex flight traits such as maneuvering and hovering. Considering that crickets typically fly only briefly for a major dispersal event that does not require a high level of precision, the functional consequences of wing asymmetry may be inconsequential. Applying phylogenetic comparative methods to examine variation in the wing FA of insects using distinctly unique flight patterns (e.g., hovering over flowers, aerial predation, long-distance dispersal) would elucidate whether wing symmetry is more canalized in species requiring greater flight precision.

Our prediction that FA would be lower in traits that are functionally important to each polyphenic group was not supported. Although we did find significant differences in FA in several traits, only a few have plausible functional explanations. For example, the observed FA was generally lower in segments of the hindleg relative to the foreleg. Given that all morphs rely on their hindlegs for quick terrestrial escape via jumping, it makes sense that these important locomotory structures would demonstrate greater symmetry than the non-saltorial forelegs. Additionally, we found that relative to other groups, SW males had the highest levels of FA in the tympanum—an important structure that females use to localize calling males, and that LW individuals use to avoid aerial predation by bats [64]. The tympanum may be less functionally important in SW males; therefore, its symmetry might be less developmentally constrained, much like how the auditory neurons of SW individuals are less sensitive to the ultrasound signals which are indicative of bat predation [65]. Interestingly, although predicted differences in hindwing FA were not significant, observed patterns of DA in both pairs of wings did match our functional predictions. As stated previously, we suspect that levels of FA observed in the hindwing are too low to impact performance; however, we found that for all traits with significant DA, the variation attributed to DA was 10–486-fold greater than FA (mean DA:FA ± SD = 129 ± 168; see Table 1, where DA:FA = MS_Side_/MS_Side_
_× Id_). Considering that DA at these levels would be more likely to have an impact on flight performance, it is worth noting that DA in the hindwing was only twofold greater than FA and that the hindwing was the only trait for which DA was not statistically significant in any of the four groups. Further highlighting the unique absence of DA in hindwings, the only other univariate trait that did not exhibit significant DA was the hind femur of SW males. Additionally, because only males produce acoustic signals by stridulating the right forewing over the left [66], forewings exhibited significant DA in males but not females. To fully understand the functional implications of asymmetry on discrete traits, our findings demonstrate that the presence and degree of both FA and DA need to be taken into consideration. Despite the fact that many of our predictions were not supported, all of our results for both the forewings and hindwings are consistent with their functional uses, suggesting either that (1) asymmetry in wings is highly controlled, or more generally, that (2) in cases where there is only one pair of structures to perform a function (e.g., one pair of flight wings vs. three pairs of legs), mechanisms controlling asymmetry might be more tightly controlled.

To the best of our knowledge, only one other study of a polyphenic species has compared FA in differentially important traits across morphs. Crespi and Vanderkist [30] found that in gall thrips (*Oncothrips tepperi*), vestigial wings of non-flighted soldiers had higher levels of FA than flight wings of dispersing individuals. We too predicted that, across morphs, more functionally important traits would have lower FA; however, similar patterns in our results were less apparent. There are a number of genetic and environmental factors known to impact levels of FA that could explain these differences between our findings. Crespi and Vanderkist [30] analyzed a wild population, whereas our study used individuals from two distinct colonies that were selectively bred for each morph. Evidence from *Drosophila* wings suggests that inbreeding leads to higher levels of FA [67] (Carter et al. 2009); as such, it is possible that inbreeding within captive colonies could lead to increased FA across morphs, which masks adaptive differences between groups. Relative to a study on *Gryllus bimaculatus* and *Gryllodes sigillatus* that measured FA in the hind femur, the magnitudes of our raw R–L differences for this trait were roughly tenfold greater than what they reported [68], indicating that levels of FA may differ significantly in longstanding captive populations. Second, morphological adaptations in wild populations are produced and maintained by natural selection. Our crickets, however, were derived from populations artificially selected for the LW or SW morph, not the underlying functionality of the morph-specific traits. This lack of selection on trait performance may have relaxed constraints opposing increases in the FA of traits vital to morph life history. FA is often correlated with individual quality, where individuals with greater energy stores exhibit greater symmetry [68]. In wild populations, individual differences in food acquisition likely lead to variable levels of FA; however, because crickets in our study were provided with food ad libitum, it makes sense that we did not find a relationship between body quality and FA. Collectively, we suggest that future studies investigating FA in polyphenic species should incorporate more variable food regimes that mimic more natural conditions.

## 5. Conclusions

Direct selection on genetically determined polymorphisms is likely to create the strongest form–function relationship; however, polyphenic species also exhibit environmental interactions. In such cases, alleles at polyphenic loci are integral to morph determination, but so too are a variety of environmental factors which include nutrition, population density, tactile cues, infection, and injury [69]. Even if selection does favor lower FA in specific traits, developmental factors regulating polymorphisms may impose constraints on the ability of selection to create the expected relationships between form and function. For example, when the sensitive period for morph determination occurs late in development, there is less time for distinct patterns in FA to manifest between the morphs. Much of our current understanding of form–function relationships is based primarily on fixed genetic differences between species, although we suggest that such patterns may not hold for phenotypically different morphs within a species. In summary, clear patterns identified in one polyphenic species may not be consistent with that of another if differences in natural histories alter the importance of trait function, compensation strategies minimize the impact of variability in a trait’s structure, or developmental patterns exist that limit the optimization between form and function.

## Figures and Tables

**Figure 1 insects-13-00640-f001:**
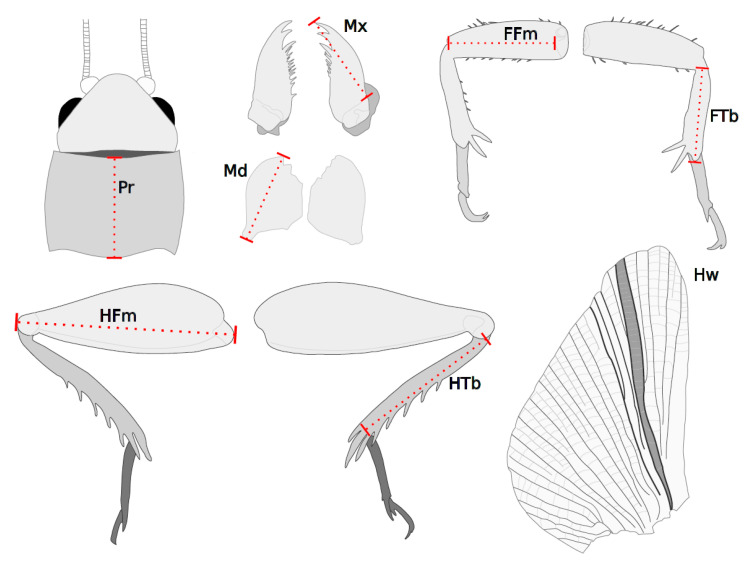
A diagram of the eight linear measurements recorded for each cricket. Illustrations depict the orientation of images from which measurements were collected: dorsal view of the head where Pr is the pronotum; ventral view of the left and right mouthparts where Mx is the maxilla and Md is the mandible; medial view of the forelimbs where FFm is the front femur and FTb is the front tibia; lateral view of the hindlimbs where HFm is the hind femur and HTb is the hind tibia; and dorsal view of a hindwing (Hw). Morphological measurements for each structure are shown as dotted lines. Drawings are not to the same scale.

**Figure 2 insects-13-00640-f002:**
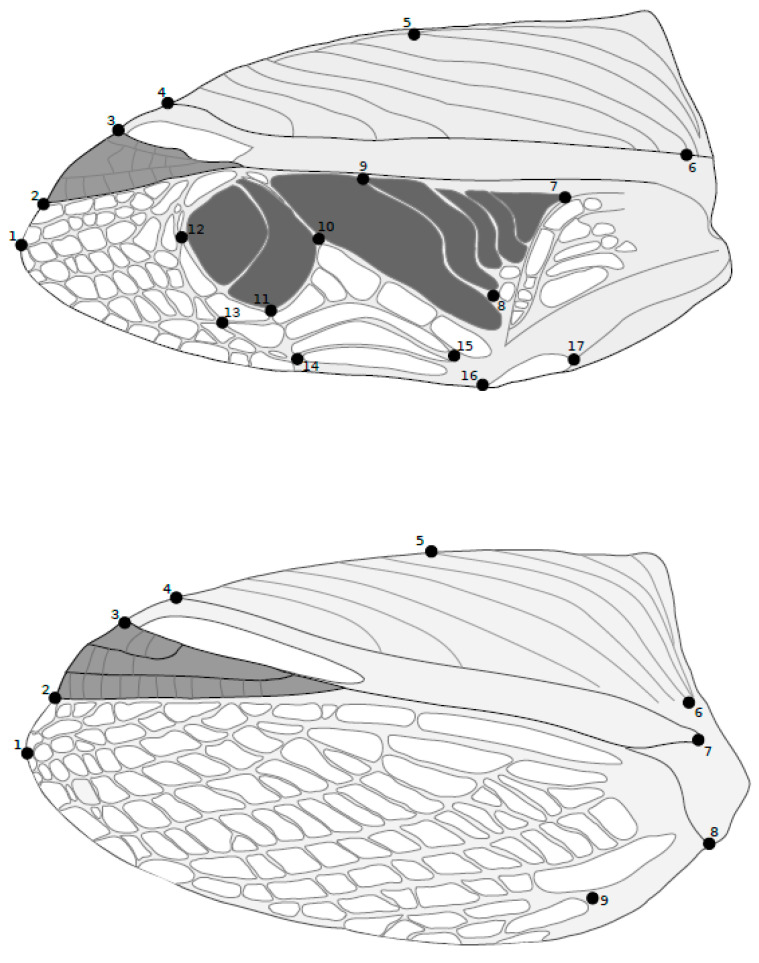
Location of the morphological landmarks on the forewing of males (**top**) and females (**bottom**). Forewings of the longwing and shortwing morphs were analyzed for each sex using geometric morphometric analyses.

**Figure 3 insects-13-00640-f003:**
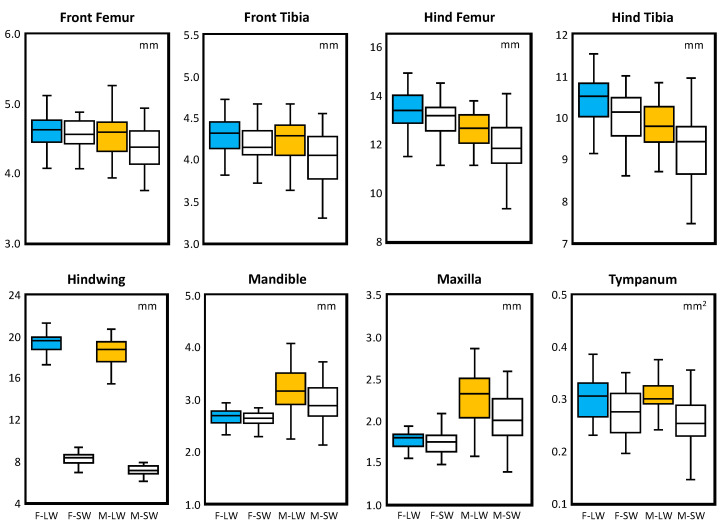
Boxplots comparing size across the eight univariate traits. Females and males are represented by blue and orange, respectively, whereas longwing and shortwing morphs are distinguished by dark and light shading. Plots of linear measurements are in mm; tympanum area is in mm^2^. Boxes enclose the median (centerline) and the 25th and 75th percentiles. Whiskers indicate the 10th and 90th percentiles. Significance levels of Tukey tests are indicated in Appendix A.

**Figure 4 insects-13-00640-f004:**
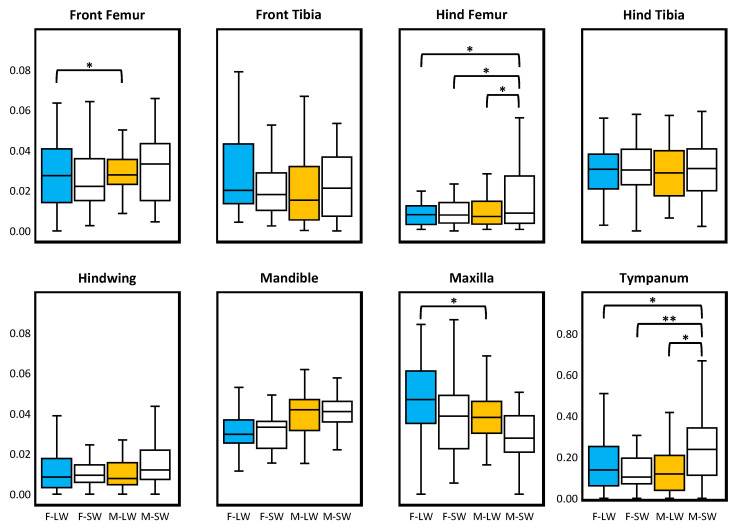
Boxplots comparing size-corrected fluctuating asymmetry values across the eight univariate traits. Females and males are represented by blue and orange, respectively, whereas longwing and shortwing morphs are distinguished by dark and light shading. Plots of linear measurements have a standardized range of 0.00–0.10 mm; tympanum has a scale of 0.0–1.0 mm^2^. Boxes enclose the median (centerline) and the 25th and 75th percentiles. Whiskers indicate the 10th and 90th percentiles. Significance levels of Tukey tests are indicated in plots: * *p* < 0.05, ** *p* < 0.01. Only significant pairwise comparisons are noted; all Tukey tests results are provided in Appendix A.

**Table 1 insects-13-00640-t001:** Results of mixed-model two-factor ANOVAs. MS refers to ‘mean sum of squares’ for left-left comparison (Sides), individuals (Id), and their interaction term (Side*Id).

Traits	N	MS_Id_	MS_Side_	MS_Side*Id_	MS_Error_	% ME	FA10
** *Front Femur* **							
Female Longwing	45	0.388 ***	0.861 ***	0.017 ***	<0.001	4.1	0.0054
Female Shortwing	31	0.395 ***	0.520 ***	0.008 ***	<0.001	12.9	0.0023
Male Longwing	39	0.507 ***	1.036 ***	0.005 ***	<0.001	16.3	0.0014
Male Shortwing	30	0.672 ***	0.844 ***	0.009 ***	<0.001	8.1	0.0028
** *Front Tibia* **							
Female Longwing	44	0.339 ***	0.559 ***	0.020 ***	<0.001	4.1	0.0064
Female Shortwing	32	0.270 ***	0.229 ***	0.011 ***	<0.001	5.9	0.0034
Male Longwing	38	0.409 ***	0.170 **	0.016 ***	<0.001	4.8	0.0051
Male Shortwing	29	0.613 ***	0.238 ***	0.010 ***	<0.001	7.4	0.0031
** *Hind Femur* **							
Female Longwing	42	3.881 ***	0.505 ***	0.032 ***	<0.001	1.0	0.0106
Female Shortwing	34	4.202 ***	0.257 **	0.024 ***	<0.001	1.2	0.0079
Male Longwing	36	3.089 ***	0.262 **	0.025 ***	<0.001	1.0	0.0082
Male Shortwing	30	6.794 ***	0.359	0.115 ***	<0.001	0.2	0.0382
** *Hind Tibia* **							
Female Longwing	41	2.675 ***	5.765 ***	0.046 ***	0.009	19.8	0.0123
Female Shortwing	33	2.412 ***	4.337 ***	0.033 ***	0.003	9.7	0.0099
Male Longwing	38	1.937 ***	4.272 ***	0.044 ***	0.009	20.5	0.0117
Male Shortwing	30	3.768 ***	3.219 ***	0.057 ***	0.002	4.1	0.0182
** *Hindwing* **							
Female Longwing	41	7.514 ***	0.059	0.198 ***	0.014	7.1	0.0613
Female Shortwing	33	2.114 ***	0.140	0.040 ***	0.003	7.5	0.0123
Male Longwing	33	9.431 ***	0.053	0.091 ***	0.014	15.4	0.0257
Male Shortwing	30	2.729 ***	0.147	0.039 ***	0.003	7.7	0.0120
** *Mandible* **							
Female Longwing	47	0.143 ***	0.496 ***	0.001 ***	<0.001	1.8	0.0003
Female Shortwing	32	0.159 ***	0.308 ***	0.001 ***	<0.001	2.5	0.0002
Male Longwing	40	0.828 ***	0.970 ***	0.003 ***	<0.001	0.7	0.0009
Male Shortwing	33	0.890 ***	0.686 ***	0.001 ***	<0.001	1.0	0.0004
** *Maxilla* **							
Female Longwing	46	0.066 ***	0.451 ***	0.002 ***	<0.001	0.8	0.0006
Female Shortwing	35	0.100 ***	0.170 ***	0.003 ***	<0.001	2.7	0.0011
Male Longwing	41	0.469 ***	0.471 ***	0.002 ***	<0.001	0.9	0.0005
Male Shortwing	32	0.486 ***	0.168 ***	0.002 ***	<0.001	0.5	0.0007
** *Tympanum* **							
Female Longwing	45	0.007 *	0.088 ***	0.004 ***	<0.001	7.2	0.0012
Female Shortwing	31	0.009 ***	0.037 ***	0.003 ***	<0.001	9.4	0.0008
Male Longwing	39	0.007 **	0.055 ***	0.003 ***	<0.001	8.7	0.0009
Male Shortwing	30	0.009	0.088 **	0.009 ***	<0.001	2.2	0.0029

% ME: % measurement error = (MSError/MSSide*Id) × 100 = ME3 [49]; FA10 = (MSSide*Id − MSError)/M, where M is the number of measurements [18,48]; Significant results are indicated by: * <0.05; ** <0.01; *** <0.001.

**Table 2 insects-13-00640-t002:** Descriptive statistics of morphological variables. Significance of DA determined using a one-sample *t*-test and indicated by bolded values.

Traits	Sample Size	Size (mm)	DA		Corrected FA
	N	Mean ± SD	Value	*p*	Mean ± SD
** *Front Femur* **					
Female Longwing	45	04.58 ± 0.25	−0.0249	**<0.001**	0.117 ± 0.059
Female Shortwing	31	04.55 ± 0.26	−0.0232	**<0.001**	0.104 ± 0.043
Male Longwing	39	04.53 ± 0.29	−0.0294	**<0.001**	0.087 ± 0.043
Male Shortwing	30	04.32 ± 0.33	−0.0317	**<0.001**	0.107 ± 0.051
** *Front Tibia* **					
Female Longwing	44	04.28 ± 0.24	0.0213	**<0.001**	0.122 ± 0.070
Female Shortwing	32	04.17 ± 0.21	0.0163	**<0.001**	0.106 ± 0.058
Male Longwing	38	04.25 ± 0.26	0.0126	**0.002**	0.120 ± 0.058
Male Shortwing	29	04.02 ± 0.32	0.0188	**<0.001**	0.118 ± 0.052
** *Hind Femur* **					
Female Longwing	42	13.38 ± 0.80	−0.0067	**<0.001**	0.082 ± 0.038
Female Shortwing	34	12.98 ± 0.84	−0.0053	**0.004**	0.080 ± 0.035
Male Longwing	36	12.61 ± 0.72	−0.0053	**0.004**	0.079 ± 0.039
Male Shortwing	30	11.91 ± 1.06	−0.0074	0.100	0.111 ± 0.064
** *Hind Tibia* **					
Female Longwing	41	10.43 ± 0.67	−0.0297	**<0.001**	0.097 ± 0.052
Female Shortwing	33	09.99 ± 0.63	−0.0296	**<0.001**	0.088 ± 0.048
Male Longwing	38	09.82 ± 0.57	−0.0278	**<0.001**	0.099 ± 0.054
Male Shortwing	30	09.22 ± 0.79	−0.0292	**<0.001**	0.105 ± 0.063
** *Hindwing* **					
Female Longwing	41	19.26 ± 1.12	0.0015	0.620	0.101 ± 0.055
Female Shortwing	33	08.22 ± 0.59	−0.0066	0.057	0.109 ± 0.050
Male Longwing	33	18.47 ± 1.25	−0.0018	0.444	0.098 ± 0.037
Male Shortwing	30	07.12 ± 0.67	−0.0084	0.057	0.114 ± 0.063
** *Mandible* **					
Female Longwing	47	2.65 ± 0.15	−0.0313	**<0.001**	0.077 ± 0.032
Female Shortwing	32	2.59 ± 0.16	−0.0313	**<0.001**	0.080 ± 0.031
Male Longwing	40	3.15 ± 0.37	−0.0401	**<0.001**	0.086 ± 0.035
Male Shortwing	33	2.89 ± 0.39	−0.0408	**<0.001**	0.075 ± 0.031
** *Maxilla* **					
Female Longwing	46	1.75 ± 0.11	−0.0471	**<0.001**	0.116 ± 0.052
Female Shortwing	35	1.72 ± 0.13	−0.0320	**<0.001**	0.126 ± 0.061
Male Longwing	41	2.25 ± 0.28	−0.0386	**<0.001**	0.092 ± 0.042
Male Shortwing	32	2.01 ± 0.29	−0.0287	**<0.001**	0.099 ± 0.048
** *Tympanum* ** ***					
Female Longwing	45	0.30 ± 0.04	0.1189	**<0.001**	0.347 ± 0.146
Female Shortwing	31	0.27 ± 0.04	0.1056	**<0.001**	0.305 ± 0.146
Male Longwing	39	0.30 ± 0.03	0.1056	**<0.001**	0.298 ± 0.148
Male Shortwing	30	0.25 ± 0.04	0.1677	**0.004**	0.447 ± 0.181

* Measured in mm^2^.

## Data Availability

The data presented in this study will be openly available in Dryad upon acceptance of this article for publication.

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
