# Peer review of "Fluctuating Asymmetry in the Polymorphic Sand Cricket (Gryllus firmus): Are More Functionally Important Structures Always More Symmetric?"

_insects, 2022, doi:10.3390/insects13070640_

Round 1
Reviewer 1 Report
The paper by Whalen et al. measures and analyzes asymmetry in four categories of cricket morphs combining hindwing length and sex. The rationale for the study seems clear and there is not a lot of information about FA in polymorphic animals, as the authors bring up. I have not seen an FA study in some time, and this work offers an alternative viewpoint to the popular FA studies of 20 years ago, which were mostly using FA to measure resilience to developmental hardships.
Abstract
The Abstract is mostly clear. An exception is the directional asymmetry result in line 35 that comes out of nowhere. Perhaps mention that DA was measured as a component of FA in the experimental analysis design.
Introduction
DA must be defined with the accompanying abbreviation in the Intro. Directional assymetry first appears in line 102 in regards to a published study without definition or abbreviated as DA.
How were traits most vital for maximizing fitness assertained? These traits should be introduced and enumarated with references for their importance in fitness.
Methods
Reference and justification for using pronotum length as a size measure? There is some disagreement about the best measure of size for crickets, see Weissman & Gray 2019 Zootaxa.
Forewings are properly termed tegmina in Orthoptera. The correct morphological terminology is preferable; however the authors may have decided to use a more generic term for a general audience.
Ambitious mesaurement scheme!
Statistics seem carefully thought out.
Results
Line 323 linear regression was a post-hoc test? The purpose of which was to verify the size standardization procedure? Introduction and explanation of this test in Methods is preferable.
Line 339 what a peculiar result that the left structures were nearly always more well developed!
Not surprising that male forewings are strongly asymmetrical: right tegmen is always over left tegmen in Gryllidae, the functional stridulatory file is on the right and the scraper on the left.
Discussion
Crickets are not particularly good fliers, and long winged crickets do not typically fly for exceptionally long distances or long durations. The Fernandez paper is for Sphingidae which are long-lived strong fliers. Later saw that this was addressed in Pg. 3.
Some long-winged crickets have deciduous hindwing that are shed after the dispersal flight. Is this the case in G. firmus? Having disposable wings may mean that minimizing FA is not a terribly important consideration in their design.
Some figures missing (Fig X), Pg. 2.
Props to you all for reporting negative results! I feel strongly that negative results should be published, especially from a study stuch as this with exhaustive corrections and considerable sample size.
Line 429 about the tympanum in SW males being less important, males may listen in on other males and may phonotax to them to aggregate or adopt satellite strategies. Tthe function of the tympana may remain important, and symmetrical tympana may improve male competitive ability.
How much genetic variation is in the colony? Are they inbred? This was considered in Pg. 5. This may be an important takeaway message that could appear in the abstract. This could also explain quirks like the left structures being more well developed in these colonies.
Reviewer 2 Report
Dear authors,
My main concern with the reviewed paper is the use of a long-term laboratory colony for comparing two morphs, with the assumption that the individuals of each morph in the lab colony exhibit the same characteristics as they would in a purely natural setting. The long-winged morph is expected to invest more in dispersal by flight, but will only reproduce later, while the short-winged morph will reproduce much earlier. The used laboratory colony originated in 1995 (so probably over 20 years at the time the experiments were set up), and has been artificially split in two lines that were selected for the presence of respectively the long-winged and the short-winged morph. Due to this visual selection, it is unclear to me whether the two morphs still differ in the original, biological differences, like the ability to fly long distances, or their reproduction. The results shown in this paper show that the hind wing length is still different between both morph, but as this is the only trait for which the two lines were actively selected, this is to be expected. I am very curious as to whether the other traits, like the ability to fly and the presence of increased wing musculature in long-winged morphs, or the enhanced reproduction in short-winged morphs, is still present in this laboratory population. As the premise of the whole paper is that this is indeed the case, I think the authors should provide some additional evidence on the differences between both morphs under long-term lab colony-conditions (either generated by themselves or already published elsewhere). Additionally, it would be nice to add an estimate of how many generations these crickets have been in a laboratory setting at the onset of the experiment.
Apart from this major concern, I think the paper reads very well and looks promising. I hope you will be able to address the above concern, as I do think this is a very interesting study overall, even though you did not obtain the expected results. I have listed below my other remarks:
Line 131: you state here you use both pronotum length and body fat percentage as a measure for individual quality, but you never really use the pronotum length to this aim. However, it does not seem like the pronotum length is really used in the paper, apart from its measurement and comparison between groups. I think it would be nice to indeed use the pronotum length as a measure for individual quality, as suggested here and similar to what you did for body fat percentage.
Line 147: As already mentioned above, I think this paragraph needs a little bit of additional information to better understand how inbred these crickets are expected to be. When were the experiments performed? Approximately how many generations has this cricket been in a lab setting? What is the approximate population size they are kept at in the lab?
Line 187: Figure is upside down in my version of the paper.
Line 309: “Body size (i.e. pronotum length) …” Body size and pronotum length are not the same, even though this sentence makes it sound that way. Please replace with “Pronotum length, used as a proxy for body size,…” or similar.
Line 309-310: Figure 3 does not seem to contain pronotum length information. Can this information please be added?
Line 325-328: As you claim that your size-adjustment successfully removed significant size-effects, could you add the results of a similar regression analysis with the non-adjusted FA data to table S2, so that the reader can also see the effect of the size-adjustment? Also do you have an explanation for the one comparison for which there still is a significant size effect after this normalization?
Line 332: I am slightly concerned about the large range in measurement error for your measurements, especially as there even seems to be a large variation within some very similar measurements (e.g. %ME for Hind Tibia of SW-M is 20.5, but %ME for Hind Tibia of LW-M is 4.1%). I was wondering whether there is a logical explanation for this wide range of errors?
Line 350: Table 1: is there a way to visualize this data in a graph, and move this table to the Supplementary material? In addition, could you add to the table legend what the abbreviaton MS stands for?
Line 360: Table 2: Is there an error on the DA values? Could you please make the way the P-values are written consistent, and preferably without using ‘E’? Also please add which statistical test was used to obtain the P-values.
Line 366: “indicated that all four groups” -> “indicated that for all four groups”
Line 368: I believe the presence of “(Side)” here means there is a difference in the side of the wing, but not the center where the sound production happens. Could you please confirm whether this is correct?
Line 366-377: Could you please try to visualize this result? I think it could be very informative to the readers to have the average right wing and average left wing laid on top of each other, so that the difference is easily visible.
Line 380-381, line 400-405: I think the differences between both morphs in hindwing FA are very minimal, based on both the shown graph (Figure 4), and the smallest P-value of a Tukey HSD test is 0.401, found between shortwinged and longwinged males (Table S3). This P-value is not even close to reaching significance, and as such I am not convinced you can even really talk about a trend here.
Line 399-401: Please fill in the appropriate figure number(s).
Line 435: I am not sure which table you are referring to with Table R3, or for which group or comparison you are listing the mean and SD?
Line 449: You could extend this logic to the hind legs which are used for jumping, and also seem to exhibit less FA then the front leg.
Line 463-465: The paper of Mallard and Barnard in fact uses Gryllus bimaculatus and Gryllodes sigillatus, so not two Gryllus species as noted here. Additionally, in that study, both species were also kept in laboratory colonies, but the authors suggest that a difference in inbreeding is causing the difference in raw L-R differences in both studies. Could the authors elaborate on why they think there is a difference in inbreeding in both studies?
Reviewer 3 Report
Summary: The asymmetry of bilateral structures is a recent niche speciality in evolutionary biology, and research has been aimed at determining its relation to developmental processes,its functional consequences (if any)n, and the effect of these on selection. Where a polymorphism such as asymmetry is genetically determined, selection is likely to create strong form/function relationships, and this has been demonstrated in various organisms. The present study now aims to explore the existence of such relationships in a polyphenic species, where the polymorphism is probably influenced by both genetic and environmental factors.
The authors have studied a unique laboratory population of field crickets. For some 20 years this population has been subject to divergent selection into two lines: long-winged and short-winged morphs (ie., flight-capable and flightless adult insects. On the basis of previous studies, the authors erected a series of hypotheses and predictions about what structures should be expected to display asymmetry, and to what extent, in the two different morphs. In brief, their results fail to substantiate their hypotheses and their predictions are unfulfilled. The authors discuss these results, and plausibly attribute them in part to the absence of natural selection (as opposed to artifical selection) acting on their populations, and on the complex origins of intraspecific polyphenism
Detail:
The introduction is excellent, reviewing the relevant literature and explaining the origins of current asymmetry theory.
The final paragraph of the introduction spells out their predictions and hypotheses. allowing the reader to evaluate the results with unusual clarity.
Methods. Procedures are clearly described and seem entirely adequate for their purposes.
The Statistical analysis is described in detail, but could be easier to read. The algebraic definitions of fluctuating and directional asymmetry (the core of the paper!)are not transparent, largely because the authors use unexplained conventions in their terminology. For example, there are critical differences between their (R-L), (R-L), and |R-L|(see lines 221-226), and these are not immediately apparent.
This reviewer is only a user of statistics, not a statistican, but the procedures described seem appropriate in general - especially commendable is the use of the square root transform (Line 257) to produce a normally distributed sample.
The Discussion is arguably the most interesting part of the paper. The authors are agreeably honest, writing (line 424) “Our prediction that Fluctuating Asymmetry would be lower in traits that are functionally important to each polyphenic group was not supported”. (In this connection, the reviewer was struck by the odd phraseology used in Line 378: ”Alternative to our predictions, we found no statistically significant evidence …”. Is this an Americanism? I would have expected “Contrary to our predictions”).
This reviewer is not convinced that the morphology of the tympanal aperture is an adequate proxy for its physiological function, as the authors imply (line 430). The authors make the very good point that their experimental populations are selectively inbred, and not a wild panmyctic population, and the effect of inbreeding on asymmetry is unknown. It may very well alter normal regulatory processes in development.
The authors final conclusion (Line 487) is “while much of our understanding of form/function relationships is based primarily on fixed genetic differences between species, we suggest that such patterns may not hold for phenotypically different morphs within a species”. This is a potentially useful contribution to this field, and possibly a warning to future researchers.
In summary, this is a clearly presented account of a technically demanding study of a previously neglected facet of the field, and I recommend it for publication.
Reviewer 4 Report
Comments of reviewer
Manuscript needs minor corrections.
1. Page 1, line 9
Simple Summary: Asymmetry ...
Summary must be bold.
2. Page 1, line 40
Add in Keywords important information:
cricket; Orthoptera; Gryllidae, Gryllus
3. Page 3, line 111
Add to species name the author and year of description, i.e. replace
‘field cricket, Gryllus firmus. Field crickets, including ....’
for ‘field cricket, Gryllus firmus Scudder, 1902. Field crickets, including ....’
4. Page 5, Fig. 1
This figure is given incorrectly. Turn up a figure!
5. Page 15, lines 399 and 401
In both cases (Fig X) are cited. I don’t find such figure in MS. Add the correct number of this figure.
6. Page 16, lines 435-436.
Authors sited Table R3. I don’t find such table in MS. Probably it is Table S3, see Supplementary Materials on page 17. Clarify.
7. Page 16, line 463
‘... study of two related Gryllus species that measured FA ...’
Add such related species in text as follow:
‘... study of two related Gryllus species (Gryllus bimaculatus and Gryllodes sigillatus) that measured FA ...’
Round 2
Reviewer 2 Report
I would like to thank the authors for their excellent work on this manuscript, and I think it is now sufficient for publication. I have only two minor comments, as noted below.
Line 120-122: I don't believe it is a very common mechanism in Orthoptera to loose their wings after dispersion. Would it be possible to rephrase the sentence so that it either just focuses on the mechanism in Gryllus without any reference to other orthopterans, or so that it narrows down the groups in which dealating actually happens rather than referring to Orthoptera in general?
Line 136: as the tympanum is also involved in hearing sounds, is not also involved in reproductive investment?
